# Therapeutic Efficacy of Sesquiterpene Farnesol in Treatment of *Cutibacterium acnes*-Induced Dermal Disorders

**DOI:** 10.3390/molecules26185723

**Published:** 2021-09-21

**Authors:** Guan-Xuan Wu, Yu-Wen Wang, Chun-Shien Wu, Yen-Hung Lin, Chih-Hsin Hung, Han-Hsiang Huang, Shyh-Ming Kuo

**Affiliations:** 1Department of Biomedical Engineering, I-Shou University, Kaohsiung City 82445, Taiwan; turtle01311995@gmail.com (G.-X.W.); yuwen870928@gmail.com (Y.-W.W.); 2Center of General Education, I-Shou University, Kaohsiung City 82445, Taiwan; wucs@isu.edu.tw; 3Institute of Biotechnology and Chemical Engineering, I-Shou University, Kaohsiung City 82445, Taiwan; glen0212@gmail.com (Y.-H.L.); chhung@isu.edu.tw (C.-H.H.); 4Department of Pharmacy and Master Program, Tajen University, Yanpu Township 90741, Taiwan; 5Department of Veterinary Medicine, National Chiayi University, Chiayi City 60054, Taiwan; hhuang@mail.ncyu.edu.tw

**Keywords:** farnesol, *Cutibacterium acnes*, clindamycin, antibacterial, anti-inflammatory

## Abstract

Acne vulgaris is a highly prevalent skin disorder requiring treatment and management by dermatologists. Antibiotics such as clindamycin are commonly used to treat acne vulgaris. However, from both medical and public health perspectives, the development of alternative remedies has become essential due to the increase in antibiotic resistance. Topical therapy is useful as a single or combined treatment for mild and moderate acne and is often employed as maintenance therapy. Thus, the current study investigated the anti-inflammatory, antibacterial, and restorative effects of sesquiterpene farnesol on acne vulgaris induced by *Cutibacterium acnes* (*C. acnes*) in vitro and in a rat model. The minimum inhibitory concentration (MIC) of farnesol against *C. acnes* was 0.14 mM, and the IC_50_ of 24 h exposure to farnesol in HaCaT keratinocytes was approximately 1.4 mM. Moreover, 0.8 mM farnesol exhibited the strongest effects in terms of the alleviation of inflammatory responses and abscesses and necrotic tissue repair in *C.*
*acnes*-induced acne lesions; 0.4 mM farnesol and clindamycin gel also exerted similar actions after a two-time treatment. By contrast, nearly doubling the tissue repair scores, 0.4 mM farnesol displayed great anti-inflammatory and the strongest reparative actions after a four-time treatment, followed by 0.8 mM farnesol and a commercial gel. Approximately 2–10-fold decreases in interleukin (IL)-1β, IL-6, and tumor necrosis factor (TNF)-α, found by Western blot analysis, were predominantly consistent with the histopathological findings and tissue repair scores. The basal hydroxypropyl methylcellulose (HPMC) gel did not exert anti-inflammatory or reparative effects on rat acne lesions. Our results suggest that the topical application of a gel containing farnesol is a promising alternative remedy for acne vulgaris.

## 1. Introduction

Acne, also known as acne vulgaris, is a chronic inflammatory skin disorder that is caused by factors such as excess sebum production, hormonal imbalance, inflammation, and external bacterial infection [1,2]. Hormones, specifically androgens, appear to be part of the mechanism underlying increased sebum production. Another common factor is the excessive growth of the bacterium *Cutibacterium acnes*, which is present on the skin [3]. *C.*
*acnes* is a gram-positive, anaerobic, lipophilic bacterium and is often found in sebaceous follicles located on the face, chest, and back of the majority of humans [4]. *C. acnes* is involved in the development of inflammatory acne by activating complements and metabolizing sebaceous triglycerides into fatty acids that irritate the follicular wall and dermis nearby [2]. The typical pathogenesis features of acne include the excess secretion of oily sebum by the skin and hyperplasia of the sebaceous glands, microcomedones, comedones, inflammatory papules, pustules, and nodules [5]. Proliferation of *C. acnes* may contribute to inflammation by activating Toll-like receptor 2, which in turn increases the release of proinflammatory cytokines, such as tumor necrosis factor (TNF)-α, interleukin (IL)-β, IL-6, and IL-8 [6].

Treatments for acne, including antibiotics, comedolytic agents, and anti-inflammatory drugs, are available as topical therapies. Treatment commonly involves applying the drug, such as clindamycin, benzoyl peroxide, or salicylic acid, directly to the affected skin [2]. However, the excessive use of these drugs for an extended period can lead to drug resistance in bacteria. Moreover, the occurrence of side effects, such as headache, skin dryness, and nausea, limits the usage of these drugs. To overcome these limitations, many alternatives, such as effective and safe antiacne drugs developed from herbal resources, have been pursued. Studies have mainly used aromatic plants as the basis of target drugs due to their common antibacterial properties [2,7].

Farnesol is a sesquiterpene alcohol possessing anti-inflammatory properties; it also has potential use as an antimicrobial agent to inhibit the growth of some microorganisms, such as the human pathogen *Staphylococcus aureus* [8]. The crucial role that inflammation may play in the development and progression of acne has been increasingly recognized. Our previous studies have indicated the novel therapeutic effects of different farnesol dosages on the skin. Liposomal or pure farnesol protected against ultraviolet B (UVB) radiation, increased collagen production, and improved skin smoothness as well as exerted anti-inflammatory and tissue-repairing effects on UVB-induced sunburn and third-degree burns in both the epidermis (in vitro) and dermis (in vivo) [9,10]. The *C. acnes*-induced inflammatory response is involved in the development of acne vulgaris; this spurred our interest in examining the potential anti-inflammatory effects of farnesol on *C. acnes*-induced inflammatory reactions and the subsequent antiacne effects on acne vulgaris. Therefore, in the current study, a rat model of dorsal skin with acne vulgaris was established through the intradermal injection of *C. acnes*. The effects of farnesol and a commercial clindamycin gel on acne vulgaris in vivo were examined by performing histopathological analyses and Western blotting. The findings of the present study can be helpful for the development of natural acne remedies in the future.

## 2. Materials and Methods

### 2.1. Materials and Animals

Farnesol was purchased from Sigma (CAS No: 4602-84-0, St. Louis, MO, USA). Commercial clindamycin gel (10 mg/g) was obtained from China Chemical and Pharmaceutical Corporation (Taipei, Taiwan)*. C. acnes* was purchased from the Bioresource Collection and Research Center. Commercial products for acne treatment were purchased from CBC Biotechnological and Pharmaceutical (No: 10723, Taiwan). Each gram of the gel contained 10 mg of clindamycin. Hydroxypropyl methyl cellulose (HPMC) was purchased from Sigma (St. Louis, MO, USA). All reagents used in this present study were of reagent grade. 

Animal experiments conducted in this study were approved by the Institutional Animal Care and Use Committee of I-Shou University, Taiwan (IACUC-ISU-107-022, approval date: 12 December 2018). 

### 2.2. Growth Curve of C. acnes

A fresh single colony of *C. acnes* was inoculated into 5 mL of reinforced clostridial medium (RCM) (NEOGEN, Sejong, Korea) mixed with 1% glucose and cultured at 37 °C for 16 h in an anaerobic environment created by Mitsubishi AnaeroPack-Anaero (Mitsubishi gas chemical Company, Tokyo, Japan) anaerobic gas generator in an anaerobic environment. Fresh *C. acnes* culture was added to a 250 mL flask with 30 mL of RCM to a final concentration of OD_600_ = 0.01 and incubated at 37 °C. 1 mL of the bacterial culture was removed at 4 h intervals to determine the bacterial concentration with the values of OD_600_ and colony-forming units (CFUs) per milliliter. 

### 2.3. Antimicrobial Effect of Farnesol on C. acnes

A fresh single colony of *C. acnes* was inoculated into 5 mL of RCM mixed with 1% glucose and cultured in an anaerobic cultivation environment at 37 °C. Two-fold serial dilutions of farnesol (100 μL) with methanol were added to a 96-well plate at 1.15 to 0.008 mM concentrations in accordance with the National Committee for Clinical Laboratory Standards (NCCLS 2020). Subsequently, 100 μL of fresh *C. acnes* culture with OD_600_ = 0.1 was added to each farnesol dilution (100 μL) to obtain the final concentrations of 0.576, 0.288, 0.144, 0.072, 0.036, 0.018, 0.009, and 0.004 mM in each well. The 96-well plate was covered with a lid and incubated at 37 °C in an anaerobic environment. Bacterial optical densities were evaluated at intervals over a 24 h period by using a microplate reader (Model 4300, Awareness Technology, FL, USA) at a wavelength of 600 nm. The minimum inhibitory concentration (MIC) was defined as the lowest concentration of farnesol that inhibited the growth of *C. acnes*.

### 2.4. In Vitro Cell Cytotoxicity Assay for Farnesol

HaCaT keratinocytes were suspended at a density of 7 × 10^3^ cells/mL and seeded onto 96-well plates. Various concentrations of farnesol were added to each well in triplicate. The cell viability of HaCaT keratinocytes was examined using the 3-(4,5-dimethylthiazol-2-yl)-2,5-diphenyltetrazolium bromide (MTT) assay. After 24 h exposure to farnesol, 20 μL of the MTT solution (5 mg/mL) was added to each well, and the cells were incubated for an additional 3 h. The formed formazan precipitate was dissolved in 200 μL of dimethyl sulfoxide, and the solution was vigorously mixed to dissolve the dye. Absorbance was measured at 570 nm by using a multiplate reader (Thermo Fisher Scientific, Waltham, MA, USA).

### 2.5. In Vivo Animal Experiments: Establishment of Skin Acne Model in a Rat

A total of 30 Sprague–Dawley male rats were infected with rat skin acne induced by *C. acnes*. The dorsal skin of the rats was intradermally inoculated with 800 μL of broth solution containing 5.0 ± 0.8 × 10^8^ CFU/mL *C. acnes* (in nine spots) [11]. An additional 500 μL of broth solution containing 5.0 ± 0.8 × 10^8^ CFU/mL *C. acnes* was intradermally inoculated the next day. The rats were randomly assigned into the experimental group (gel-based treatment; *n* = 24), untreated group without any treatment (*n* = 3), or normal group (without *C. acnes*, *n* = 3). All the rats had access to normal food and water after acne induction. The four experimental groups were as follows: Group A rats were treated with 0.4 mM farnesol gel, Group B rats were treated with 0.8 mM farnesol gel, Group C rats were treated with commercial clindamycin gel, and Group D rats were treated with pure gel. The gel was prepared by dissolving HPMC (2%) in 90 °C distilled water. The solution was cooled to room temperature, and HA (0.5%) and xanthan gum (0.5%) were added and mixed through stirring. Subsequently, farnesol was added and stirred at 100 rpm for 10 min to prepare the gel with farnesol. 

In vivo animal experiments were performed over 9 days (Figure 1). The first 2-day period was reserved for the intradermal injection of *C. acnes*, and the following 2-day period for the development of acne vulgaris on the skin. The rats were treated with pure gel, clindamycin gel, and gel containing 0.4 or 0.8 mM (one treatment/day) farnesol for 2 days or 4 days. Subsequently, the rats were sacrificed, and their skin samples were harvested for the histopathological analysis of skin repair properties. The rats receiving the intradermal injection of bacterial broth were anesthetized using zoletil (tiletamine with zolazepam, 40 mg/kg, i.p., 50 mg/kg, i.p., respectively) and xylazine (10 mg/kg, i.p.). During the study period, clinical signs of pain, salivation, and abnormal behavior were carefully monitored.

### 2.6. Histopathological Analyses

After a 2-day or 4-day treatment with pure gel, gel containing farnesol, or commercial gel, the rat skin was harvested, and the sample was fixed in 10% neutral-buffered formalin. The skin samples were then dehydrated in graded ethanol solutions, cleared in xylene, embedded in paraffin blocks, and cut into 5 μm thick sections. Hematoxylin and eosin (H&E) staining was performed to histopathologically examine the rat skin. Furthermore, Masson’s trichrome staining was performed to examine collagen changes in the skin of the rats treated with pure gel, gel containing farnesol, or commercial gel. ImageJ software (version 1.50; National Institute of Health, USA) was used to obtain the semiquantitative measurements of the collagen content in each group. The color settings in ImageJ remained unchanged throughout the analysis of blue-stained areas in each sample. Samples were evaluated at 100× magnification, and the calculation was repeated in three microscopic fields.

### 2.7. Histopathological Scoring of Skin Repair

To evaluate the anti-inflammatory and reparative effects of farnesol on *C. acnes*-induced lesions in the rat skin, the level of epithelization, regeneration, and reparation of pilosebaceous units or epithelial cysts; alleviation of inflammatory cell infiltration; improvement of abscesses and necrotic tissues; and collagenization in the rat skin were assessed.

### 2.8. Western Blot Analysis

To determine the anti-inflammatory and reparative effects of farnesol on rat skin acne, the rat skin specimens were harvested after a 2- or 4-day treatment. The samples were placed in cooled phosphate-buffered solution (pH 7.2). Each sample was ground until it was fully homogenized. Ice-cold lysis buffer was added to lyse the tissue for 1 h. Thereafter, the tissue was centrifuged at 13,000 rpm at 4 °C for 15 min. Proteins were separated through 10% sodium dodecyl sulfate–polyacrylamide gel electrophoresis and subsequently transferred and blotted onto a nitrocellulose membrane. Nonspecific binding sites on the membrane were blocked using 5% skim milk powder in Tris-buffered saline with 0.05% Tween 20 for 1 h at room temperature. The membrane was then incubated with primary antimouse IL-1β, IL-6, TNF-α, and α-tubulin antibodies (Santa Cruz Biotechnology, Santa Cruz, CA, USA) overnight at 4 °C. The protein levels of IL-1β, IL-6, and TNF-α were compared between the treatment and control groups by performing a semiquantitative intensity analysis (normalized by the respective α-tubulin and background) with ImageJ software.

### 2.9. Statistical Analysis

All values were expressed as means ± standard errors of the mean. The results were analyzed through one-way analysis of variance (ANOVA) by using SPSS version 20.0 to determine significant differences among experimental groups (*p* < 0.05).

## 3. Results and Discussion

### 3.1. Growth of C. acnes and MIC Determination

The *C. acnes* exhibited a slow growth rate; the log phase began at 4 h and proceeded until the stationary phase began at 24 h (data not shown). The optical density (OD_600_, i.e., the absorption at a wavelength of 600 nm) was 0.01 ± 0.0017 at 4 h and 3.16 ± 0.07 at 24 h.

In the assessment of the MIC of farnesol against *C. acnes*, the broth microdilution sensitivity of *C. acnes* was measured with increasing concentrations of farnesol in vitro. In the broth assay, inhibitory effects were noted when farnesol at concentrations of approximately 0.56, 0.28, and 0.14 mM prevented the change of OD_600_ values between 0 and 48 h (Figure 2). The data suggested that the MIC of farnesol was 0.14 mM, and this can be used as the reference setting for farnesol doses in the future in vivo trials. The antibacterial effects herein were similar to those shown in the previous investigations, suggesting that farnesol possesses antimicrobial capacity against various pathogens [12,13].

### 3.2. In Vitro Viability of Keratinocytes after Farnesol Treatment

The results of the MTT assay showed that farnesol significantly inhibited the viability of keratinocytes at a higher concentration of 2 mM (Figure 3A). The IC_50_ of farnesol for HaCaT human keratinocytes was approximately 1.4 mM after 24 h exposure. The cell morphology results revealed that treatment with 2 mM farnesol led to the loss of the normal spindle-like shape of keratinocytes in vitro (Figure 3B). These results indicated that 2 mM farnesol inhibited the viability of keratinocytes. In addition, our previous study indicated that farnesol at concentrations higher than 0.4 mM reduced the viability of skin fibroblasts. Thus, on the basis of the aforementioned findings, the doses of farnesol were set at 0.4 and 0.8 mM in the current study.

### 3.3. Histopathological Analysis

The intact pilosebaceous units in the dermis and epidermis of the normal rats were observed. Regular arrangement of collagen fibers was noted in the dermis. Distribution of a slight to moderate amount of the fat tissue and connective tissue was observed in the subcutaneous layer but not in the dermis. However, on the 8th day after first acne induction, clear abscesses, necrotic tissues, and inflammatory cell infiltration were found in the dermis and subcutis of the group treated twice. In addition, numerous epithelial and follicular cysts as well as typical acne lesions and erythematous papules or pustules were observed in the dermis and epidermis of these rats (Figure 4A and Figure 5A).

In the group that received 0.4 mM farnesol twice, some normal structures and reparative processes of pilosebaceous units were observed. The epithelial and follicular cysts were slightly decreased but remained discernible in the epidermis. The necrotic areas and inflammatory cells in the dermis and subcutis were decreased. In addition, regular arrangement of collagen was observed in the dermis. These results suggest that 0.4 mM farnesol exerted moderate restorative effects on the pilosebaceous units, dermis, and subcutis (Figure 4A and Figure 5A). However, the presence of epithelial and follicular cysts and the moderate number of cutaneous abscesses imply that the anti-inflammatory and antibacterial effects of 0.4 mM farnesol are suboptimal (Figure 4A). These findings were further verified by determining the tissue repair scores of the two-time treatment groups (Table 1) [14].

The epithelial cysts, necrotic tissues, and inflammatory cells were largely decreased on day 8 after acne induction in the groups that received 0.8 mM farnesol twice. Repair of pilosebaceous units was observed in the dermis and epidermis, and regular arrangement of collagen fibers was noted in the dermis (Figure 4A and Figure 5A and Table 1). These findings suggest that two-time treatment with 0.8 mM farnesol exerted moderate anti-inflammatory, antibacterial, and reparative effects on the rat acne model.

Notably, the group treated with the commercial clindamycin gel exhibited similar features as did the group treated twice with 0.8 mM farnesol. Specifically, the necrosis and abscesses in the dermis and subcutis were largely decreased, and the regular arrangement of collagen fibers in the dermis was noted (Figure 4A and Figure 5A). The only exception was that the number of epithelial cysts was slightly to moderately decreased, suggesting weaker repair of pilosebaceous units compared with that in the group treated twice with 0.8 mM farnesol. Inflammatory cells were moderately decreased in the treatment group compared with the untreated group (Figure 4A). These findings indicate that the commercial clindamycin gel could alleviate acne-induced inflammation and necrosis in the skin in a shorter period (within 7 days) but lacked the capacity to restore the pilosebaceous units. 

In the group treated twice with the HPMC gel, the histopathologic findings on day 8 revealed that the large areas of subcutaneous and dermal abscesses, inflammatory cell infiltration, and numerous epithelial cysts remained; hence, the basal solution of farnesol exhibited limited restorative and negligible anti-inflammatory effects. Nonetheless, minor repair of pilosebaceous units was observed, suggesting that HPMC moderately benefitted hair follicle repair (Figure 4A). These findings also demonstrated that farnesol was primarily responsible for the reparative, anti-inflammatory, and antibacterial actions in the skin and HPMC was merely a complementary ingredient in the current study’s gel recipe.

By contrast, in the untreated group of the four times treated category, on the 10th day after first acne induction, the abscesses, necrotic tissues, and inflammatory cell infiltration remained discernible in the dermis and subcutis (Figure 4B). However, some normal structure of pilosebaceous units could be found in the dermis and epidermis, suggesting that a physiological process of pilosebaceous repair was progressing normally at that time. Additionally, a slight to moderate decrease in epithelial cysts was observed, implying normal restoration of the dermis and epidermis on day 10 (Figure 4B).

In the group treated four times with 0.4 mM farnesol, a normal structure of pilosebaceous units and regular collagen arrangement were observed in the dermis and epidermis at day 10 after initial acne induction (Figure 4B and Figure 5B). Moreover, the dermal and subcutaneous abscesses and necrosis as well as epithelial cysts were largely decreased (Figure 4B), indicating that four treatments with 0.4 mM farnesol could restore the tissue damage and alleviate inflammatory responses caused by *C. acnes* as well as promote repair in the affected skin by day 10.

The group treated four times with 0.8 mM farnesol showed histopathologic changes similar to those of the group treated four times with 0.4 mM farnesol on day 10 after the first acne induction. Specifically, the abscesses and necrotic areas were notably decreased, and collagen arrangement was generally regular (Figure 4B and Figure 5B). Some epithelial cysts and immature pilosebaceous units were observed in the group treated four times with 0.8 mM farnesol (Figure 4B). This result differed from those of the untreated and 0.4 mM farnesol-treated groups, which could be due to the fact that longer-term treatment with a higher dose of farnesol inhibits the normal processes of hair follicles and epidermal repair. In addition, these data accord with our previous findings, in which farnesol or liposome-encapsulated treatment accelerated tissue restoration and alleviated inflammatory responses in UVB-induced sunburn and PM_2.5_-caused in vivo impairment in the epidermis and dermis [10,14,15]. Moreover, as the in vitro MIC of farnesol against *C. acnes* (0.14 mM) was lower than the farnesol doses (0.4 and 0.8 mM) used in the animal experiments in vivo in the current study, it is explainable and sensible that significant alleviation of inflammatory responses and obvious amelioration of abscesses and necrosis were found in the farnesol-treated rats in both categories.

Similar to the histopathology of the four-time 0.4 mM farnesol treatment group, the dermal and subcutaneous abscesses and necrosis as well as epithelial cysts were noticeably decreased, and most collagen was regularly arranged in the commercial clindamycin gel treatment group. However, increased numbers of immature pilosebaceous units were found in the dermis of the four-time commercial clindamycin gel treatment group on day 10, indicating that long-term topical administration of clindamycin gel could have inhibited the regeneration of pilosebaceous units and hair follicles in the rat acne model (Figure 4B). Notably, the findings of the quantitative analysis of the blue staining areas of collagen through Masson’s trichrome staining (Figure 5C,D) accord with the histopathological findings and tissue repair scores in both experimental categories, thus confirming that farnesol promotes in vivo collagen production in the rat acne model.

In the four-time HPMC gel treatment group, slight to moderate increases in epithelial cysts and immature hair follicle structures were observed on day 10. Moreover, inflammatory responses, necrosis, and abscesses were noted in the dermis and subcutaneous layer (Figure 4B and Table 1). These findings suggested that only HPMC gel is unsuitable as a therapy for acne and the long-term use of basal gel may increase moisture and reduce dryness on the skin’s surface, hindering normal skin repair processes and delaying recovery from acne.

### 3.4. Western Blot Analysis

IL-1β mRNA and the active, processed form of IL-1β are abundant in inflammatory acne lesions, and IL-1β is a key proinflammatory cytokine that has been induced by *C. acnes* both in vitro and in vivo in a murine model [16]. The findings of Western blotting in this study were largely consistent with both the present histopathological findings and those reported by Kistowska et al. [17]; that is, the untreated and basal gel-treated groups in both the two- and four-time administrative categories exhibited the highest IL-1β levels (Figure 6). Moreover, the strongest inflammatory responses were observed in these two groups (in both categories) in terms of tissue repair scores (Table 1). In addition, the findings of moderate to optimal alleviation of inflammation observed in the 0.4 and 0.8 mM farnesol as well as commercial clindamycin gel treatment groups (in both administrative categories) were also largely consistent with the Western blot results of those groups (Figure 6), demonstrating that IL-1β is an essential proinflammatory cytokine in acne lesions induced by *C. acnes* in vivo.

The crucial but possibly diverse roles of TNF-α have been revealed in our previous findings and the investigations of other researchers [13,17]. Specifically, TNF-α may play different roles in the earlier and later phases of wound healing in the skin [18]. TNF-α is pivotal in the early process of wound healing in the skin, a fact confirmed by our Western blot data of the two-time administrative category groups. Our histopathological analysis suggested the alleviation of inflammatory responses, regeneration and repair of pilosebaceous units and epithelial cysts, and improvement of abscesses and necrotic tissues in the 0.4 and 0.8 mM farnesol treatment groups (Figure 4A). However, the TNF-α proteins were slightly increased in these groups (Figure 6A), indicating a beneficial role of TNF-α in the first 2 days of farnesol treatment. Commercial clindamycin gel, by contrast, might not have induced TNF-α actions during the early treatment phase (Figure 6A). Additionally, in the four-time treatment group, both 0.4 and 0.8 mM farnesol as well as the commercial clindamycin gel considerably reduced TNF-α levels (Figure 6B), suggesting that on day 10 (after four-time treatment), TNF-α may act as a direct “master regulator” of inflammatory cytokine production, as previously reported [19]. 

IL-6 is produced at inflammatory sites and plays an essential role in the acute stage response as defined by various clinical and biological features, such as the production of acute phase proteins [19]. Furthermore, IL-6 stimulates T and B cells, thereby encouraging chronic inflammatory responses [19]. Our current Western blot and histopathological results indicating that *C. acnes*-induced inflammation and inflammatory responses were alleviated after the application of farnesol and the commercial clindamycin gel are in accordance with the findings of previous studies (Figure 4 and Figure 6). Neither untreated nor basal HPMC gel treatment group showed severe and clear inflammation with white blood cell infiltration, and the abscesses and necrotic tissues in the rat skin barely improved (Figure 4). These histopathological features and tissue repair scores (Table 1) were fully reflected by Western blot data for IL-6 (Figure 6), in which the untreated and basal HPMC gel treatment groups exhibited the highest levels of IL-6 proteins in the skin. In addition, the amelioration of inflammatory responses, abscesses, and necrotic areas after two- (acute or subchronic) and four-time farnesol or commercial clindamycin gel treatments (more chronic) observed in histopathological findings (Figure 4 and Figure 5) were consistent with the corresponding IL-6 Western blot data (Figure 6), suggesting that IL-6 is a potentially feasible indicator of recovery from *C. acnes*-induced inflammation, necrosis, and lesions in the rat skin model.

In summary, the antibacterial, anti-inflammatory, and restorative actions of farnesol were revealed in the present study. The antibacterial activity of farnesol against *C.*
*acnes* with an MIC of 0.14 mM was identified. The IC_50_ of farnesol in human keratinocytes was approximately 1.4 mM, and the doses 0.4 mM and 0.8 mM farnesol used in the current study caused less than 10% keratinocyte death in vitro. Furthermore, in the rat skin model, 0.8 mM and 0.4 mM farnesol exhibited the strongest anti-inflammatory and repair-accelerating effects in response to *C. acnes*-induced lesions in the two- and four-time treatment groups. The tissue repair activity of the farnesol gel was slightly greater than that of the commercial clindamycin gel in both treatment groups, indicating that farnesol is an effective therapy for *C. acnes*-induced dermal inflammation and necrosis in vivo. The results of the current study suggest that farnesol can be a promising remedy for acne in the future.

## Figures and Tables

**Figure 1 molecules-26-05723-f001:**
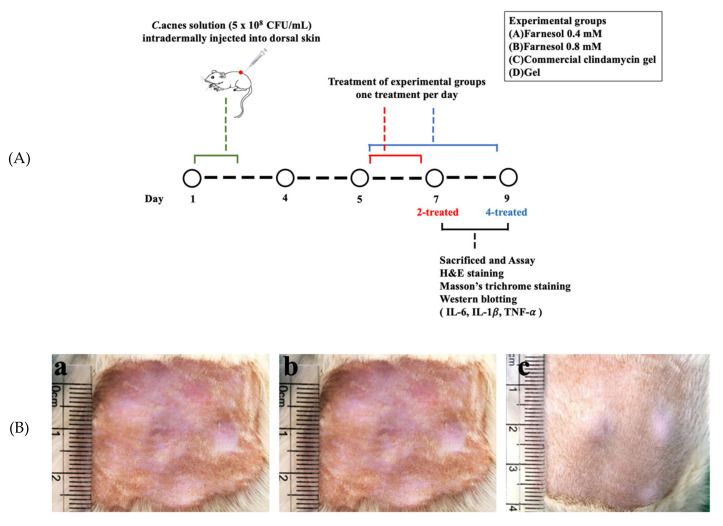
(**A**) Schematic of the development and the time scale of treatment of *C. acnes*-induced rat skin acne. (**B**) The dorsal skin of a rat injected with 5 × 10^8^ CFU/mL *C. acnes* at 96 h postinjection (**a**), 144 h postinjection (**b**), and 192 h postinjection (**c**).

**Figure 2 molecules-26-05723-f002:**
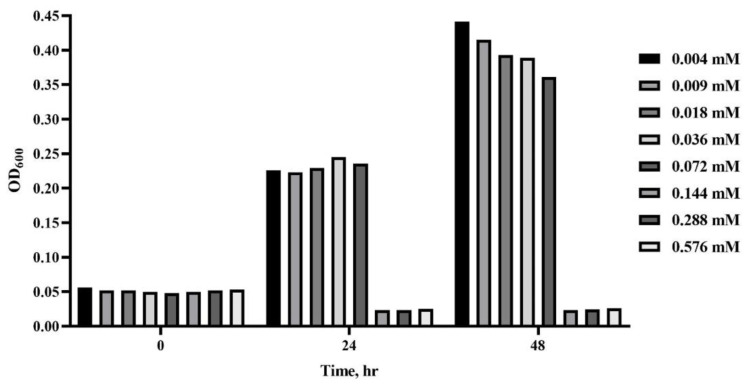
MIC test results for farnesol in the treatment of *C.*
*acnes*-induced acne after 24 and 48 h incubations. The MIC was measured by assessing the effects of a series of farnesol concentrations (from 0.004 to 0.576 mM) on the absorbance at 600 nm (OD_600_). The MIC was determined as 0.14 mM.

**Figure 3 molecules-26-05723-f003:**
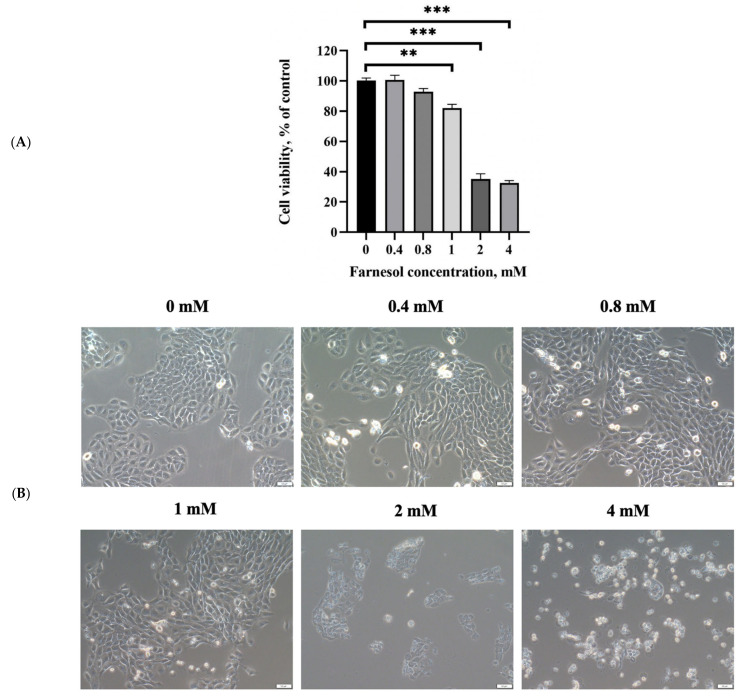
(**A**) Cell viability assay displaying the low cytotoxicity of farnesol on HaCaT keratinocytes. (**B**) Cell morphology images of keratinocytes after various treatments with the indicated the concentrations of farnesol, supporting the cell viability assay results (Bar, 50 μm). ** *p* < 0.01; *** *p* < 0.001 (through ANOVA).

**Figure 4 molecules-26-05723-f004:**
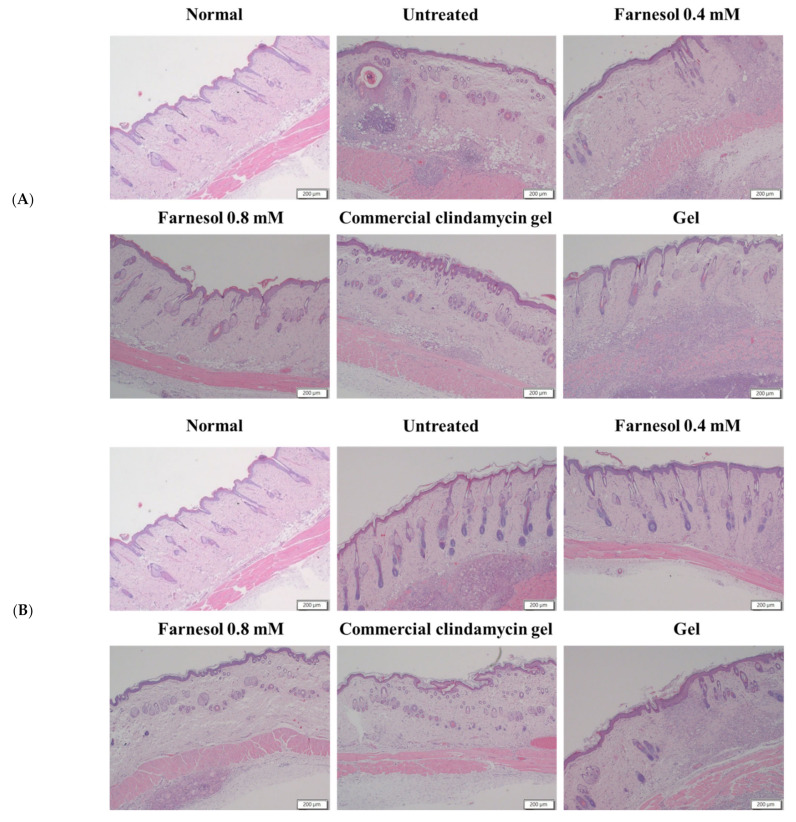
H&E stains of the dorsal skin of rat after two-time (**A**) and four-time (**B**) treatment with 0.4 mM farnesol, 0.8 mM farnesol, HPMC, and commercial clindamycin gels.

**Figure 5 molecules-26-05723-f005:**
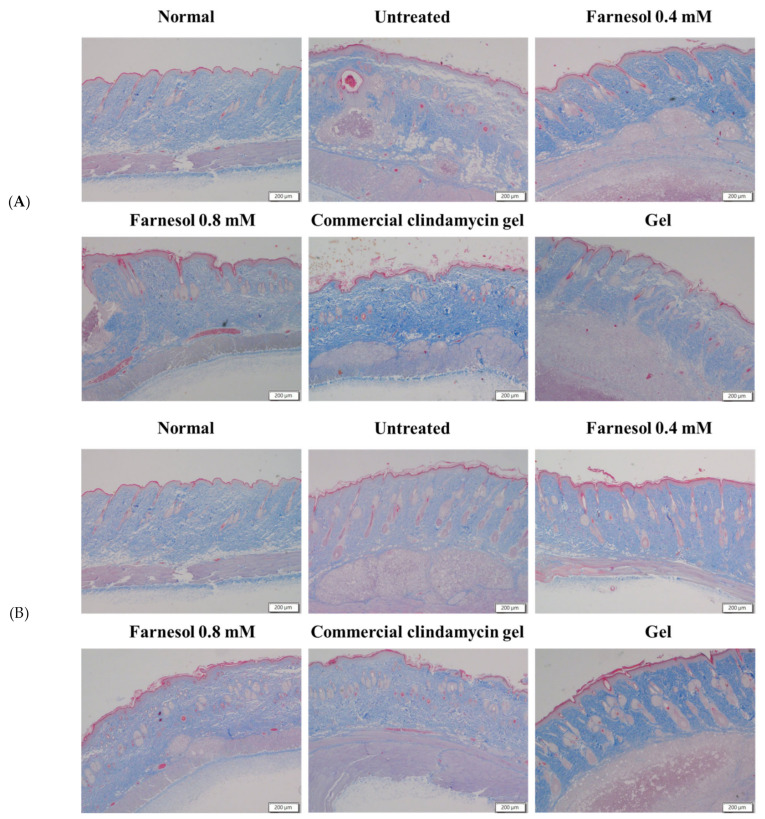
Masson’s trichrome staining of the dorsal skin of rats after two-time (**A**) and four-time (**B**) treatment with 0.4 mM farnesol, 0.8 mM farnesol, HPMC, and commercial clindamycin gels. Semiquantitative assessment of collagen from blue color stains in Masson’s trichrome images for (**C**) two-time and (**D**) four-time treatment groups (*n* = 3). ** *p* < 0.01 (compared with the normal group); ^#^
*p* < 0.05 (compared with the untreated group), ^##^
*p* < 0.01.

**Figure 6 molecules-26-05723-f006:**
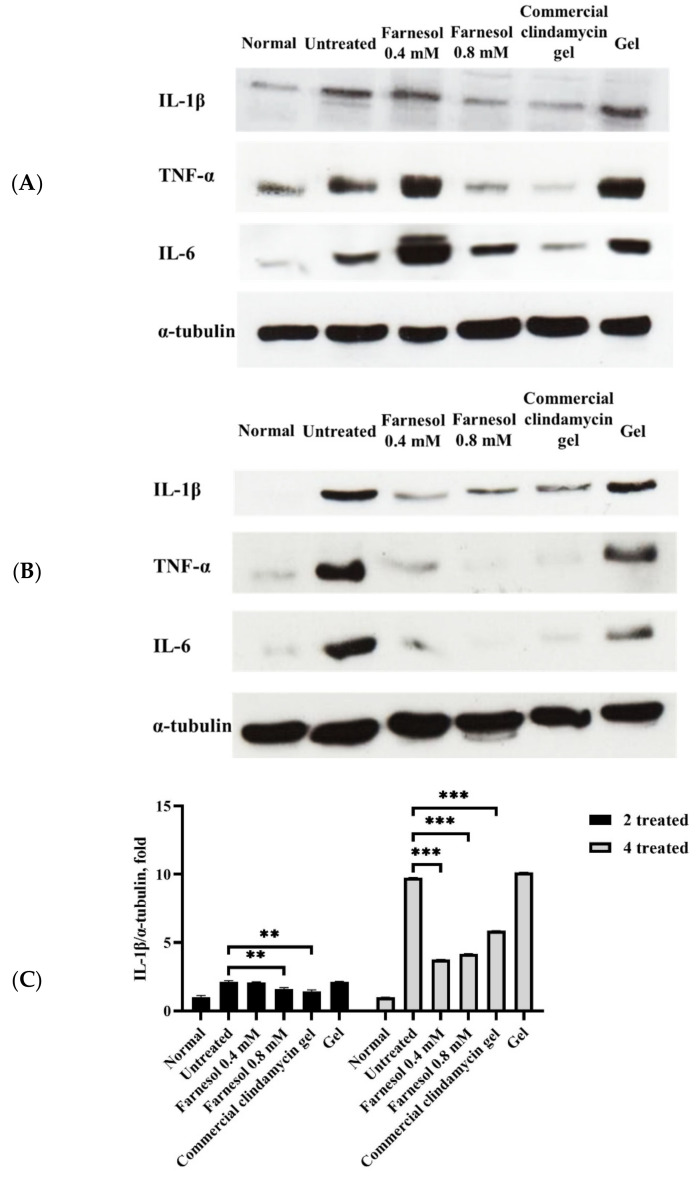
Western blot of the dorsal skin of rats after two-time (**A**) and four-time (**B**) treatment with 0.4 mM farnesol, 0.8 mM farnesol, HPMC, and commercial clindamycin gels. (**C**) Semiquantitative analyses of Western blot of proinflammatory cytokines, using the ImageJ software. * *p* < 0.05; ** *p* < 0.01; *** *p* < 0.001 (compared with the untreated group).

**Table 1 molecules-26-05723-t001:** Tissue repair scores for the epidermis and dermis of the rats that received *C. acnes* and were treated using farnesol, commercial antibiotic gel, or were untreated in the two- and four-time treatment groups.

**1. Results for the two-time treatment category (at day 8 after acne induction).**					
	A	B	C	D	E
Epithelialization	0.67 ± 0.33	2.67 ± 0.33 **	2.67 ± 0.33 **	2.00 ± 0.33	1.67 ± 0.33
Regeneration and reparation of pilosebaceous unit/epithelial cysts	0.33 ± 0.33	3.33 ± 0.33 ***^,#^	2.33 ± 0.33 *	1.33 ± 0.33	1.67 ± 0.33
Alleviation of inflammatory cell infiltration	0.33 ± 0.33	3.67 ± 0.33 ***^,###^	4.33 ± 0.33 ***^,###^	3.33 ± 0.33 ***^,###^	0.33 ± 0.33
Improvement of abscesses and necrotic tissues	0.33 ± 0.33	3.67 ± 0.33 ***^,###^	4.33 ± 0.33 ***^,###^	4.00 ± 0.00 ***^,###^	0.33 ± 0.33
Collagenization	1.33 ± 0.33	4.33 ± 0.33 ***^,###^	4.33 ± 0.33 ***^,###^	4.00 ± 0.33 ***^,##^	1.67 ± 0.33
Total	3.00 ± 0.58	17.67 ± 0.88 ***^,###^	18.00 ± 0.58 ***^,###^	14.67 ± 0.33 ***^,###^	5.67 ± 0.33
**2. Results for the four-time treatment category (at day 10 after acne induction).**					
	A	B	C	D	E
Epithelialization	2.67 ± 0.33	3.33 ± 0.33 ^#^	3.67 ± 0.33 ^#^	3.33 ± 0.33 ^#^	1.67 ± 0.33
Regeneration and reparation of pilosebaceous unit/epithelial cysts	2.67 ± 0.33	3.67 ± 0.33 ^#^	2.33 ± 0.33	2.33 ± 0.33	1.67 ± 0.33
Alleviation of inflammatory cell infiltration	1.67 ± 0.33	4.33 ± 0.33 ***^,###^	4.67 ± 0.33 ***^,###^	4.00 ± 0.00 **^,###^	1.33 ± 0.33
Improvement of abscesses and necrotic tissues	1.67 ± 0.33	4.33 ± 0.33 **^,###^	4.33 ± 0.33 **^,###^	4.67 ± 0.33 ***^,###^	1.33 ± 0.33
Collagenization	2.67 ± 0.33	4.33 ± 0.33	3.67 ± 0.33	3.67 ± 0.33	3.33 ± 0.33
Total	11.33 ± 0.33	20.00 ± 0.58 ***^,###^	18.67 ± 0.67 ***^,###^	18.00 ± 0.58 ***^,###^	9.33 ± 0.33

Acute/subchronic/chronic inflammation: severe (0) moderate (1) mild (2) mild/none (3) none (4). Scores are expressed as means ± standard errors of means; significant differences were determined using one-way ANOVA and the Tukey–Kramer test. * *p* < 0.05; ** *p* < 0.01; *** *p* < 0.001 (compared with the untreated group) and # *p* < 0.05; ## *p* < 0.01; ### *p* < 0.001 (compared with the basal gel–treated group). Group A: untreated group (negative control 1); Group B: treated with 0.4 mM pure farnesol; Group C: treated with 0.8 mM pure farnesol; Group D: treated with HPMC gel; Group E: treated with commercial clindamycin gel (positive control).

## Data Availability

Data are available in the present study.

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
