# Peer review of "Therapeutic Efficacy of Sesquiterpene Farnesol in Treatment of Cutibacterium acnes-Induced Dermal Disorders"

_molecules, 2021, doi:10.3390/molecules26185723_

Round 1

Reviewer 1 Report

The article highlighted the efficacy of the chemical, farnesol, in the treatment of acne vulgaris and proposed it as an alternative therapy for the disease. Although the article provides good evidence through in-vivo data, there are a few issues that are overlooked in the study. For example, the antimicrobial activity of the farnesol was not illustrated in vivo. The in vitro MIC data was presented in mg/mL and the data of the rest of the in-vivo study was presented in mM. Therefore, it is not clear whether 0.8 mM farnesol would be able to achieve the antibiotic activity in vivo? Authors should discuss this in their analysis. Next, there is no direct comparison analysis between the efficacy of commercial and farnesol gel preparation as all statistical comparison was done with control or untreated group. Therefore, there is no substantial evidence that the proposed compound is superior to commercial preparation. The article should address/discuss this issue. Also, in figure 5, comparisons were made between the normal and treatment group to highlights the preferable statistical significance. However, the actual comparison should be made between the untreated and treated groups.

Some minor comments on the manuscript are

Line 24 -26: not informative in the abstract, please consider using exact values

Line 83: repeated sentence

Figure5 -6 – be consistent in using terms Control and Normal.

Author Response

Please see the attached file, thanks.

Reviewer 2 Report

The manuscript molecules-1364124 investigates the anti-inflammatory, antibacterial and restorative effects of sesquiterpene farnesol in the treatment of C. acnes-induced dermal disorders. 

Manuscript is well organized and results are interesting. I find the manuscript to be potentially worthy of publication in Molecules. However, I have some minor comments for the improvement of presented data before  publishing.

The Materials and Methods section need to be modified, at several places there is repeated information, e.g. see the lines 75 and 83, 86 and 88 (highlighted in the attached pdf file). The sentence at line 90 "Some of the bacterial culture (1 mL) was removed" need to be rewritten (why authors use "some" when it is 1 ml?)

line 109: rewrite "after 24-h exposure to treatment", either "after 24-h exposure to farnesol..."or "24-h treatment with farnesol"

lines 119-120: explain the difference between control group and normal group

lines 122-134: again, the information is repeated in these two paragraphs

Fig. 1 A: need to be modified, it is confusing (red and blue lines - both represents one treatment per day?), what does 2t and 4t mean?

page 161: add the info about pH of the buffer

My main comment concerns the concentration of the farnesol. In the experiments showing the growth of C.acnes  concentration in microgram/mL is used while in the rest of the manuscript the concentrations in mM are used. Authors should either unify the units or add th info what is the approximate molarity of solutions used in growth experiments.

Lines 238-244: do they represent a legend to the table? 

Figure 6 (C): I miss the statistical analysis of the results presented in graphs

Author Response

Please see the attached file, thanks.

Reviewer 3 Report

  1. Brief summary: Original paper, dealing with interesting topic, involved in treatment Cutibacterium acnes-induces dermal disorders in laboratory mice end efficacy of farnesol at different concentrations compared to commercial antibiotics and control without treatment.
  2. Title: is clear, it describes the topic of the article well.
  3. Abstract: is comprehensive
  4. The structure of the article is correct.
  5. Materials and methods: this section needs several additions, some methods are not clearly explained (see below).

Line 78 contains unexplained abbreviation HPMC.

Line 104 “that inhibited the growth ...” It means visible growth? Or was another method used to determine the MIC?

The inoculation solution containing 5x105 CFU/ml is described in the "In vivo animal experiments" section of line 117. It is not possible to achieve such a specific number of CFU/ml for many inoculations.  It is necessary to describe how you detected and verified this number (but rather the +/- interface) and what the real result was.

Line 124 and also throughout the article you refer to commercial gel with clindamycin as "commercial".  This designation does not describe the main property of this gel, i.e. the content of clindamycin, and in the following text and graphs this property disappears.  It is advisable to change the abbreviated label throughout the article, which clearly identifies the antibiotic content.

Line 126 contains unexplained abbreviation HA.

Histopathological analysis is described from line 142. There is no clear concept in this paragraph: “semiquantitative measurement”. How is it done, what is measured? What is the scale? In Figure 1 you have mm and cm, but in the results (figure 5C, 5D) they are %.  What is 100%? It is necessary to explain better and more precisely.

In the next method from line 159, it is again not clear how the “semiquantitative intensity” (line 170) is made.  The results (Figure 6C) do not show any units.

6.       Results:

It is advisable to add exact values to Figure 3A, maximum of cell viability 120% is unlikely.

Figure 4 and 5: explanation of the abbreviation FOH is too far from the picture and is not used throughout the article except for some figures. For a quicker understanding of the figures, it is advisable not to use or unify throughout the article.

Figure 5C and 5D: unclear figures due to ambiguity in the assessment of the methodology (see above “semiquantitative measurement”). Control column reaches only 0.5%? The vertical axis range is 0-0.8% (or 0-80%)?

Table 1:  Tissue repair scores is not defined. Where did it come from, how did it come from? What is the range, what units? What is the best value, what is the worst? It is necessary to add to the methodology. In addition, it is appropriate to add to the table the number of samples tested in each group (n).

Figure 6C:  Units on the vertical axis are not added, which again is based on an unclear methodology (see above). What does the “normal” column mean? Such a group is not defined in the testing.

Another discussion is excellent and clear.

  1. References: The article contains appropriate and adequate references to related works.

Summary of the review: Original paper, the article contains interesting facts and results, but it is necessary to add to the shortcomings in the description of the methodology and the resulting context in the results.

Author Response

Please see the attached file, thanks.

Round 2

Reviewer 1 Report

The manuscript has substantially improved. I don't have any further comments.